# Efficient Online Portfolio with Logarithmic Regret

**Haipeng Luo**
Department of Computer Science
University of Southern California
haipengl@usc.edu

**Chen-Yu Wei**
Department of Computer Science
University of Southern California
chenyu.wei@usc.edu

**Kai Zheng**
Key Laboratory of Machine Perception, MOE, School of EECS, Peking University
Center for Data Science, Peking University, Beijing Institute of Big Data Research
zhengk92@pku.edu.cn

## Abstract

We study the decades-old problem of online portfolio management and propose the first algorithm with logarithmic regret that is not based on Cover's Universal Portfolio algorithm and admits much faster implementation. Specifically Universal Portfolio enjoys optimal regret $\mathcal{O}(N \ln T)$ for $N$ financial instruments over $T$ rounds, but requires log-concave sampling and has a large polynomial running time. Our algorithm, on the other hand, ensures a slightly larger but still logarithmic regret of $\mathcal{O}(N^2 (\ln T)^4)$, and is based on the well-studied Online Mirror Descent framework with a novel regularizer that can be implemented via standard optimization methods in time $\mathcal{O}(T N^{2.5})$ per round. The regret of all other existing works is either polynomial in $T$ or has a potentially unbounded factor such as the inverse of the smallest price relative.

## 1 Introduction

We consider the well-known online portfolio management problem [8], where a learner has to sequentially decide how to allocate her wealth over a set of $N$ financial instruments in order to maximize her return, importantly under no assumptions at all on how the market behaves. Specifically, for each trading period $t = 1, \ldots, T$, the learner first decides the proportion of her wealth to invest on each stock, and then by the end of the period observes the return of each stock and continues to invest with her total wealth. The goal of the learner is to maximize the ratio between her total wealth after $T$ rounds and the total wealth of the best *constant-rebalanced portfolio* (CRP) which always rebalances the wealth to ensure a fixed proportion of investment for each stock. Equivalently, the learner aims to minimize her *regret*, which is the negative logarithm of the aforementioned ratio.

The minimax optimal regret for this problem is $\mathcal{O}(N \ln T)$, achieved by Cover's Universal Portfolio algorithm [8]. This algorithm requires sampling from a log-concave distribution and all known efficient implementations have large polynomial (in $N$ and $T$) running time, such as $\mathcal{O}(T^{14} N^4)$ [15].[1]

Online Newton Step (ONS) [12], on the other hand, follows the well-studied framework of Online Mirror Descent (OMD) with a simple time-varying regularizer and admits much faster implementation via standard optimization methods. The regret of ONS is $\mathcal{O}(GN \ln T)$ where $G$ is the largest gradient $\ell_\infty$-norm encountered over $T$ rounds (formally defined in Section 1.1) and can be arbitrarily large making the bound meaningless. A typical way to prevent unbounded gradient is to mix the output

Table 1: Comparisons of regret and running time of different algorithms. Note that $G$ is potentially unbounded. For running time, we assume Interior Point Method is used to solve the involved optimization problems, and omit all logarithmic (in $N$ and $T$) factors.

| Algorithm | Regret | Time (per round) |
|---|---|---|
| Universal Portfolio [8, 15] | $N \ln T$ | $T^{14} N^4$ |
| ONS [12] | $GN \ln T$ | $N^{3.5}$ |
| FTRL [3] | $G^2 N \ln(NT)$ | $TN^{2.5}$ |
| EG [14] | $G\sqrt{T \ln N}$ | $N$ |
| Soft-Bayes [19] | $\sqrt{NT \ln N}$ | $N$ |
| ADA-BARRONS (**this work**) | $N^2 (\ln T)^4$ | $TN^{2.5}$ |

of ONS with a small amount of uniform distribution, which after optimal trade-off can at best lead to a regret bound of $\mathcal{O}(N\sqrt{T \ln T})$. An earlier work [3] achieves a worse regret bound of $\mathcal{O}(G^2 N \ln(NT))$ via an efficient algorithm based on another well-known framework Follow-the-Regularized-Leader (FTRL).

There are also extremely efficient approaches with time complexity $\mathcal{O}(N)$ or $\mathcal{O}(N \ln N)$ per round, such as exponentiated gradient [14], online gradient descent [22], and Soft-Bayes from recent work of [19]. The first two achieve regret of order $\mathcal{O}(G\sqrt{T \ln N})$ and $\mathcal{O}(G\sqrt{T})$ respectively[2] while the last one achieves $\mathcal{O}(\sqrt{NT \ln N})$ without the dependence on $G$. Despite being highly efficient, all of these approaches fail to achieve the optimal logarithmic dependence on $T$ for the regret.

As one can see, earlier works all exhibit a trade-off between regret and time complexity. A long-standing open question is how fast an algorithm with optimal regret can be. Specifically, are there algorithms with optimal regret and similar or even better time complexity compared to ONS?

In this work, we make a significant step toward answering this question by proposing a simple algorithm with regret $\mathcal{O}(N^2 (\ln T)^4)$ and time complexity $\mathcal{O}(TN^{2.5})$ per round. To the best of our knowledge, this is the first algorithm with logarithmic regret (and no dependence on $G$) that is not based on Cover's algorithm and admits fast implementation comparable to ONS and [3]. As a comparison, we show in Table 1 the regret and time complexity of existing works and ours, where for OMD/FTRL-type algorithms we do a naive calculation of the running time based on the Interior Point Method [18] (to solve the key optimization problems involved), despite the possibility of even faster implementation.

Our algorithm is parameter-free and deterministic. It follows the OMD framework with a novel regularizer that is a mixture of the one used in ONS and the so-called log-barrier (a.k.a. Burg entropy) [10, 2, 21].[3] Critically, our algorithm also relies on an increasing learning rate schedule for the log-barrier similar to recent works on bandit problems [2, 21], as well as another sophisticated adaptive tuning method for the learning rate of the ONS regularizer, which resembles the standard doubling trick but requires new analysis since monotonicity does not hold for our problem.

## 1.1 Notation and Setup

The online portfolio problem fits into the well-studied online learning framework (see for example [13]). Formally, the problem proceeds for $T$ rounds (for some $T > N$). On each round $t = 1, \ldots, T$, the learner first decides a distribution $x_t \in \Delta_N$ where $\Delta_N$ is the $(N-1)$-dimensional simplex. After that the learner observes the *price relative vector* $r_t \in \mathbb{R}_+^N$ so that her total wealth changes by a factor of $\langle x_t, r_t \rangle$. Taking the negative logarithm, this corresponds to observing a loss function $f_t(x) = -\ln \langle x_t, r_t \rangle$ for $x \in \Delta_N$, chosen arbitrarily by an adversary. The regret of the

**Algorithm 1:** BARrier-Regularized Online Newton Step (BARRONS)

1 **Input**: $0 < \beta \leq \frac{1}{2}, 0 < \eta \leq 1$
2 **Define**: $\bar{\Delta}_N = \{x \in \Delta_N : x_i \geq \frac{1}{NT}, \; \forall i\}$
3 **Initialize**: $x_1 = \frac{1}{N}\mathbf{1}$, $A_0 = NI_N$ where $\mathbf{1}$ is the all-one vector and $I_N$ is the $N \times N$ identity matrix.
4 **for** $t = 1, 2, \ldots$ **do**
5 $\quad$ Predict $x_t$ and observe loss function $f_t(x) = -\ln \langle x, r_t \rangle$.
6 $\quad$ Make updates

$$A_t = A_{t-1} + \nabla_t \nabla_t^\top$$

$$\eta_{t,i} = \eta \exp\left(\max_{s \in [t]} \log_T \frac{1}{Nx_{s,i}}\right) \quad (1)$$

$$x_{t+1} = \operatorname*{argmin}_{x \in \bar{\Delta}_N} \langle x, \nabla_t \rangle + D_{\psi_t}(x, x_t) \quad (2)$$

$\quad$ where $\nabla_t = \nabla f_t(x_t)$ and $\psi_t(x) = \frac{\beta}{2}\|x\|_{A_t}^2 + \sum_{i=1}^N \frac{1}{\eta_{t,i}} \ln \frac{1}{x_i}$.

learner against a CRP parameterized by $u \in \Delta_N$ is then defined as

$$\text{Reg}(u) = \sum_{t=1}^T f_t(x_t) - \sum_{t=1}^T f_t(u) = -\ln \frac{\Pi_{t=1}^T \langle x_t, r_t \rangle}{\Pi_{t=1}^T \langle u, r_t \rangle},$$

which is exactly the negative logarithm of the ratio of total wealth per dollar invested between the learner and the CRP $u$. Our goal is to minimize the regret against the best CRP, that is, to minimize $\max_{u \in \Delta_N} \text{Reg}(u)$. This setup is also useful for other non-financial applications such as data compression [9, 19].

Note that the regret is invariant to the scaling of each $r_t$ and thus without loss of generality we assume $\max_{i \in [N]} r_{t,i} = 1$ for all $t$ where we use the notation $[n]$ to represent the set $\{1, \ldots, n\}$. It is now clear what the aforementioned largest gradient norm $G$ formally is: $G = \max_{t \in [T]} \|\nabla f_t(x_t)\|_\infty = \max_{t \in [T], i \in [N]} \frac{r_{t,i}}{\langle x_t, r_t \rangle} \leq \min\{\frac{1}{\min_{t,i} r_{t,i}}, \frac{1}{\min_{t,i} x_{t,i}}\}$, which in general can be unbounded. To control its magnitude, previous works [3, 4, 12, 14] either explicitly force $x_{t,i}$ to be lower bounded, which leads to worse regret, or make the so-called no-junk-bonds assumption (that is, $\min_{t,i} r_{t,i}$ is not too small), which might make sense for the portfolio problem but not other applications [19]. Our main technical contribution is to show how this term can be automatically canceled by a negative regret term obtained from the log-barrier regularizer with increasing learning rates.

## 2 Barrier-Regularized Online Newton Step

Recall that for a sequence of convex regularizers $\psi_t$, the outputs of Online Mirror Descent are defined by $x_{t+1} = \operatorname*{argmin}_{x \in \Delta_N} \langle x, \nabla_t \rangle + D_{\psi_t}(x, x_t)$ where $\nabla_t$ is a shorthand for $\nabla f_t(x_t)$, $D_{\psi_t}(x, y) = \psi_t(x) - \psi_t(y) - \langle \nabla \psi_t(y), x - y \rangle$ is the Bregman divergence associated with $\psi_t$, and we start with $x_1$ being the uniform distribution. The intuition is that we would like $x_{t+1}$ to have small loss with respect to a linear approximation of $f_t$, and at the same time to be close to the previous decision $x_t$ to ensure stability of the algorithm.

Although not presented in this form originally, Online Newton Step [12] is an instance of OMD with $\psi_t(x) = \frac{\beta}{2}\|x\|_{A_t}^2 = \frac{\beta}{2}x^\top A_t x$ where $A_t = A_{t-1} + \nabla_t \nabla_t^\top$ (for some $A_0$) is the gradient covariance matrix and $\beta$ is a parameter. The analysis of [12] shows that the regret of ONS for the portfolio problem is $\text{Reg}(u) = \mathcal{O}(\frac{N \ln T}{\beta})$ as long as $\beta \leq \min\{\frac{1}{2}, \min_{s \in [T]} \frac{1}{8|(u-x_s)^\top \nabla_s|}\}$. Even assuming an oracle tuning, by Hölder inequality this gives $\mathcal{O}(GN \ln T)$ as mentioned.

To get rid of the dependence on $G$, we observe the following. Since $\nabla_t = -\frac{r_t}{\langle x_t, r_t \rangle}$, its $\ell_\infty$-norm is large only when there is a stock with high reward $r_{t,i}$ while the learner puts a small weight $x_{t,i}$ on it. However, the reason that the weight $x_{t,i}$ is small is because the learner finds it performing poorly

prior to round $t$, which means that the learner had better choices and actually should have performed better than stock $i$ previously (that is, negative regret against stock $i$). Now as stock $i$ becomes good at time $t$ and potentially in the future, as long as the learner can pick up this change quickly, the overall regret should not be too large.

Similar observations were made in previous work [2, 21] for different problems in the bandit setting, where they introduced the log-barrier regularizer with increasing learning rate to explicitly ensure a large negative regret term based on the intuition above. This motivates us to add an extra log-barrier regularizer to ONS for our problem. Specifically, we define our regularizer to be the following mixture: $\psi_t(x) \triangleq \frac{\beta}{2} \|x\|_{A_t}^2 + \sum_{i=1}^{N} \frac{1}{\eta_{t,i}} \ln \frac{1}{x_i}$, where $\eta_{t,i}$ is individual and time-varying learning rate. Different from previous work, we propose a more adaptive tuning schedule for these learning rates based on Eq. (1) (instead of a doubling schedule [2, 21]), but the key idea is the same: increase the learning rate for a stock when its weight is small so that the algorithm learns faster in case the stock becomes better in the future. Another modification is that we force the decision set to be $\bar{\Delta}_N = \{x \in \Delta_N : x_i \geq \frac{1}{NT}, \; \forall i\}$ instead of $\Delta_N$ to ensure an explicit lower bound for $x_{t,i}$. We call this algorithm BARrier-Regularized Online Newton Step (BARRONS) (see Algorithm 1).

Under the same condition on $\beta$ as for ONS, we prove the following key theorem for BARRONS which highlights the important negative regret term obtained from the extra log-barrier regularizer. Note that it is enough to provide a regret bound only against smooth CRP $u \in \bar{\Delta}_N$ since one can verify that the total loss of any CRP $u \in \Delta_N$ can be approximated by a smooth CRP in $\bar{\Delta}_N$ up to an additive constant of 2 (Lemma 10 in Appendix B).

**Theorem 1.** *For any $u \in \bar{\Delta}_N$, if $\beta \leq \alpha_T(u)$ for $\alpha_t(u) \triangleq \min\left\{\frac{1}{2}, \min_{s \in [t]} \frac{1}{8|(u-x_s)^\top \nabla_s|}\right\}$, then* BARRONS *ensures*

$$\text{Reg}(u) \leq \mathcal{O}\left(\frac{N \ln T}{\eta}\right) + \frac{8N \ln T}{\beta} - \frac{1}{8(\ln T)\eta} \sum_{i=1}^{N} \max_{t \in [T]} \frac{u_i}{x_{t,i}}. \tag{3}$$

The second term of Eq. (3) comes from ONS while the rest comes from the log-barrier. To see why the negative term is useful, for a moment assume that we were able to pick $\beta$ such that $\frac{1}{2}\alpha_T(u^*) \leq \beta \leq \alpha_T(u^*)$ where $u^*$ is the best (smoothed) CRP. Then by setting $\eta = \frac{1}{1024N(\ln T)^2}$, the regret against $u^*$ can be upper bounded by

$$\mathcal{O}\left(\frac{N \ln T}{\eta}\right) + \frac{16N \ln T}{\alpha_T(u^*)} - \frac{1}{8(\ln T)\eta} \sum_{i=1}^{N} \max_{t \in [T]} \frac{u_i^*}{x_{t,i}}$$

$$\leq \mathcal{O}\left(\frac{N \ln T}{\eta}\right) + 128N(\ln T) \max_{t \in [T]} \left|\frac{\langle r_t, u^* - x_t \rangle}{\langle r_t, x_t \rangle}\right| + 32N \ln T - \frac{1}{8(\ln T)\eta} \sum_{i=1}^{N} \max_{t \in [T]} \frac{u_i^*}{x_{t,i}}$$

$$\leq \mathcal{O}\left(\frac{N \ln T}{\eta}\right) + 128N(\ln T) \left(\max_{t \in [T],i} \frac{u_i^*}{x_{t,i}} + 1\right) + 32N \ln T - 128N(\ln T) \sum_{i=1}^{N} \max_{t \in [T]} \frac{u_i^*}{x_{t,i}}$$

$$\leq \mathcal{O}\left(N^2(\ln T)^3\right),$$

which completely eliminates the dependence on the largest gradient norm $G$!

The problem is, of course, it is not clear at all how to tune $\beta$ in this way ahead of time. On a closer look, it is in fact not even clear whether such $\beta$ exists since $\alpha_T(u^*)$ depends on the sequence $x_1, \ldots, x_T$ and thus also on $\beta$ itself (see Appendix A for more discussions). Assuming its existence, a natural idea would be to run many copies of BARRONS with different $\beta$ and to choose them adaptively via another online learning algorithm such as Hedge [11]. We are unable to analyze this method due to some technical challenges (discussed in Appendix A), let alone the fact that it leads to much higher time complexity making the algorithm impractical. In the next section, however, we completely resolve this issue via an adaptive tunning scheme for $\beta$.

## 3 ADA-BARRONS

Our main idea to resolve the parameter tuning issue is based on a more involved doubling trick. As dicussed we would like to set $\beta$ to be roughly $\alpha_T(u_T^*)$ where $u_t^* = \min_u \sum_{s \leq t} f_s(u)$. A standard

---
**Algorithm 2:** ADA-BARRONS
---
1 **Initialize**: $\beta = \frac{1}{2}, \eta = \frac{1}{2048N(\ln T)^2}, \gamma = \frac{1}{25}$
2 Run BARRONS with parameter $\beta$ and $\eta$, where after each round $t$, if the following holds:

$$\beta > \alpha_t(u_t), \tag{4}$$

with $\alpha_t$ defined in Theorem 1 and

$$u_t = \operatorname*{argmin}_{u \in \bar{\Delta}_N} \sum_{s=1}^{t} f_s(u) + \frac{1}{\gamma} \sum_{i=1}^{N} \ln \frac{1}{u_i}, \tag{5}$$

then set $\beta \leftarrow \frac{\beta}{2}$, and rerun BARRONS from Line 2 with time index reset to 1.

---

doubling trick would suggest halving $\beta$ whenever it is larger than $\alpha_t(u_t^*)$ and then restart the algorithm. However, since $\alpha_t(u_t^*)$ is not monotone in $t$, standard analysis of doubling trick does not work.

Fortunately, due to the special structure of our problem, we are able to analyze a slight variant of the above proposal where we halve the value of $\beta$ whenever it is larger than $\alpha_t(u_t)$, for the *regularized leader* $u_t$ (defined in Eq. (5)) instead of the actual leader $u_t^*$. The regularization used here to compute $u_t$ is again the log-barrier, but the purpose of using log-barrier is simply to ensure the stability of $u_t$ as discussed later. In fact, $u_t$ is exactly the prediction of the FTRL approach of [3], up to a different value of the parameter $\gamma$. Here we only use $u_t$ to assist the tunning of $\beta$.

We call the final algorithm ADA-BARRONS (see Algorithm 2). Note that for notational simplicity, we reset the time index back to 1 at the beginning of each rerun, that is, the algorithm forgets all the previous data.

To see why this works, suppose condition (4) holds at time $t$ and triggers the restart. Then we know $\alpha_t(u_t) < \beta \leq \alpha_{t-1}(u_{t-1})$. On one hand, this implies that the condition of Theorem 1 holds at time $t - 1$ for $u_{t-1}$, so the regret bound (3) holds for $u_{t-1}$; on the other hand, this also implies that the term $\frac{N \ln T}{\beta}$ in Eq. (3) is bounded by $\frac{N \ln T}{\alpha_t(u_t)}$, which further admits an upper bound in terms of $\max_{s \in [t], i \in [N]} \frac{u_{t,i}}{x_{s,i}}$ by the same calculation shown after Theorem 1. Therefore, if we can show $\max_{s \in [t], i \in [N]} \frac{u_{t,i}}{x_{s,i}} \approx \max_{s \in [t-1], i \in [N]} \frac{u_{t-1,i}}{x_{s,i}}$, then the same cancellation will happen which leads to small regret against $u_{t-1}$ for this period. It is also not hard to see that $u_{t-1}$ will have similar total loss compared to the actual best CRP $u_{t-1}^*$, leading to the desired regret bound overall.

Indeed, we show in Appendix B that both $x_t$ and $u_t$ enjoy a certain kind of stability, which then implies the following lemma.

**Lemma 2.** *If condition* (4) *holds at time $t$, then* $\max_{s \in [t-1], i \in [N]} \frac{u_{t-1,i}}{x_{s,i}} \geq \frac{1}{2} \max_{s \in [t], i \in [N]} \frac{u_{t,i}}{x_{s,i}}$.

Call the period between two restart an *epoch* and use the notation epoch($\beta$) to indicate the epoch that runs with parameter $\beta$. We then prove the following key lemma based on the discussions above.

**Lemma 3.** *For any $u \in \bar{\Delta}_N$, if epoch($\beta$) is not the last epoch, then we have*

$$\sum_{s \in epoch(\beta)} (f_s(x_s) - f_s(u)) \leq \mathcal{O}\left(N^2 (\ln T)^3\right) - \frac{8N \ln T}{\beta};$$

*otherwise,*

$$\sum_{s \in epoch(\beta)} (f_s(x_s) - f_s(u)) \leq \mathcal{O}\left(N^2 (\ln T)^3\right) + \frac{8N \ln T}{\beta}.$$

With this key lemma, we finally prove the claimed regret bound of ADA-BARRONS.

**Theorem 4.** ADA-BARRONS *ensures* $\mathrm{Reg}(u) \leq \mathcal{O}\left(N^2 (\ln T)^4\right)$ *for any $u \in \Delta_N$.*

*Proof.* Again by Lemma 10 it suffices to consider $u \in \bar{\Delta}_N$. Let the number of epochs be $B$. When $B = 1$, the bound holds trivially by Lemma 3. Otherwise, the regret is upper bounded by

$$B \times \mathcal{O}\left(N^2(\ln T)^3\right) + \left(\sum_{b=1}^{B-1} \frac{-8}{2^{-b}} + \frac{8}{2^{-B}}\right) N \ln T = B \times \mathcal{O}(N^2(\ln T)^3) + 16N \ln T.$$

Since for any $t \in [T]$ and $u \in \bar{\Delta}_N$, $\frac{1}{\alpha_t(u)} \leq \mathcal{O}(\max_{s\in[t],i} \frac{u_{t,i}}{x_{s,i}}) \leq \mathcal{O}(NT)$, which means $\alpha_t(u) \geq \Omega(\frac{1}{NT})$, condition (4) cannot hold after $\mathcal{O}(\ln(NT))$ epochs. Therefore $B = \mathcal{O}(\ln(NT)) = \mathcal{O}(\ln T)$ (since $T > N$) and the regret bound follows. $\square$

**Computational complexity.** It is clear that the computational bottleneck of our algorithm is to solve the optimization problems defined by Eq. (2) and Eq. (5). Suppose we use Interior Point Method to solve these two problems. It takes time $\mathcal{O}\left(M\sqrt{N}\log\frac{N}{\epsilon}\right)$ to obtain $1 - \epsilon$ accuracy where $M$ is the time complexity to compute the gradient and Hessian inverse of the objective [5], which in our case is $\mathcal{O}(N^3)$ for solving $x_t$ and $\mathcal{O}(TN^2 + N^3)$ for solving $u_t$. As $T > N$, the complexity per round is therefore $\mathcal{O}(TN^{2.5})$ ignoring logarithmic factors. We note that this is only a pessimistic estimation and faster implementation is highly possible, especially for solving $u_t$ (given $u_{t-1}$) in light of the efficient implementation discussed in [1] for similar problems.

## 4 Detailed Analysis

In this section we provide the key proofs for our results.

**Analysis of BARRONS** The proof of Theorem 1 is a direct combination of the following three lemmas, where the first one is by standard OMD analysis and analysis from [12] and the proof is deferred to Appendix B.

**Lemma 5.** *Under the condition of Theorem 1,* BARRONS *ensures for any* $u \in \Delta_N$

$$\text{Reg}(u) \leq \sum_{t=1}^{T} \left(\langle \nabla_t, x_t - x_{t+1}\rangle + D_{\psi_t}(u, x_t) - D_{\psi_t}(u, x_{t+1}) - \frac{\beta}{2}\langle \nabla_t, x_t - u\rangle^2\right).$$

**Lemma 6.** BARRONS *with parameters* $\beta \leq \frac{1}{2}$ *and* $\eta \leq 1$ *guarantees*

$$\sum_{t=1}^{T} \left(D_{\psi_t}(u, x_t) - D_{\psi_t}(u, x_{t+1}) - \frac{\beta}{2}\langle \nabla_t, x_t - u\rangle^2\right) \leq \mathcal{O}\left(\frac{N\ln T}{\eta}\right) - \frac{1}{8(\ln T)\eta}\sum_{i=1}^{N}\max_{t\in[T]}\frac{u_i}{x_{t,i}}.$$

*Proof.* Define $\phi_t(x) = \frac{\beta}{2}\|x\|_{A_t}^2$, $\varphi_t(x) = \sum_{i=1}^{N} \frac{1}{\eta_{t,i}}\ln\frac{1}{x_i}$. Then $\psi_t(x) = \phi_t(x) + \varphi_t(x)$ and $D_{\psi_t} = D_{\phi_t} + D_{\varphi_t}$. Note that $D_{\phi_t}(x, y) = \frac{\beta}{2}\|x - y\|_{A_t}^2$ and $D_{\varphi_t}(x, y) = \sum_{i=1}^{N} \frac{1}{\eta_{t,i}}h\left(\frac{x_i}{y_i}\right)$ where $h(z) = z - 1 - \ln z$. For notation simplicity, we also define $\eta_{0,i} = \eta_{1,i}$ for all $i$. Now we have

$$\sum_{t=1}^{T} \left(D_{\psi_t}(u, x_t) - D_{\psi_t}(u, x_{t+1})\right) \leq D_{\psi_0}(u, x_1) + \sum_{t=1}^{T}\left(D_{\psi_t}(u, x_t) - D_{\psi_{t-1}}(u, x_t)\right)$$

$$\leq D_{\psi_0}(u, x_1) + \frac{\beta}{2}\sum_{t=1}^{T}\left(\|u - x_t\|_{A_t}^2 - \|u - x_t\|_{A_{t-1}}^2\right) + \sum_{t=1}^{T}\sum_{i=1}^{N}\left(\frac{1}{\eta_{t,i}} - \frac{1}{\eta_{t-1,i}}\right)h\left(\frac{u_i}{x_{t,i}}\right)$$

$$= D_{\psi_0}(u, x_1) + \frac{\beta}{2}\sum_{t=1}^{T}\langle \nabla_t, u - x_t\rangle^2 + \sum_{t=1}^{T}\sum_{i=1}^{N}\left(\frac{1}{\eta_{t,i}} - \frac{1}{\eta_{t-1,i}}\right)h\left(\frac{u_i}{x_{t,i}}\right)$$

$$= \mathcal{O}\left(\beta N + \frac{N\ln T}{\eta}\right) + \frac{\beta}{2}\sum_{t=1}^{T}\langle \nabla_t, u - x_t\rangle^2 + \sum_{t=2}^{T}\sum_{i=1}^{N}\left(\frac{1}{\eta_{t,i}} - \frac{1}{\eta_{t-1,i}}\right)h\left(\frac{u_i}{x_{t,i}}\right). \tag{6}$$

It remains to deal with $\sum_{t=2}^{T}\sum_{i=1}^{N}\left(\frac{1}{\eta_{t,i}} - \frac{1}{\eta_{t-1,i}}\right)h\left(\frac{u_i}{x_{t,i}}\right)$. Fix $i$, let $t = s_1, s_2, \ldots, s_M \in [2, T]$ be the rounds where $\eta_{t,i} \neq \eta_{t-1,i}$. Define $s_0 = 1$ and let $\eta^{(m)} = \eta_{s_m,i}$ and $x^{(m)} = x_{s_m,i}$ for

notational convenience. Note that by definition $\eta^{(m)} = \eta \exp(\log_T \frac{1}{Nx^{(m)}}) \le \eta \exp(\log_T T) = \eta e$. We thus have

$$\sum_{t=2}^{T} \left( \frac{1}{\eta_{t,i}} - \frac{1}{\eta_{t-1,i}} \right) h \left( \frac{u_i}{x_{t,i}} \right) = \sum_{m=1}^{M} \left( \frac{1}{\eta^{(m)}} - \frac{1}{\eta^{(m-1)}} \right) h \left( \frac{u_i}{x^{(m)}} \right)$$

$$= \sum_{m=1}^{M} \left( \frac{1 - \exp\left( \log_T \frac{x^{(m-1)}}{x^{(m)}} \right)}{\eta^{(m)}} \right) h \left( \frac{u_i}{x^{(m)}} \right) \le \sum_{m=1}^{M} \left( \frac{-\log_T \frac{x^{(m-1)}}{x^{(m)}}}{\eta^{(m)}} \right) h \left( \frac{u_i}{x^{(m)}} \right)$$

$$\le \sum_{m=1}^{M} \left( -\frac{\log_2 \frac{x^{(m-1)}}{x^{(m)}} \big/ \log_2 T}{\eta e} \right) h \left( \frac{u_i}{x^{(m)}} \right) \le \sum_{m=1}^{M} \left( -\frac{\log_2 \frac{x^{(m-1)}}{x^{(m)}}}{4(\ln T)\eta} \right) h \left( \frac{u_i}{x^{(m)}} \right).$$

We first consider the case when $x^{(M)} \le \min\left\{ \frac{1}{2N}, \frac{u_i}{2} \right\}$. Because $x^{(M)} \le \frac{1}{2N} = \frac{x^{(0)}}{2}$ and $x^{(m)}$ is decreasing in $m$, there must exist an $m^* \in \{1, 2, \dots, M\}$ such that $\frac{x^{(m^*-1)}}{x^{(M)}} \ge 2$ and $\frac{x^{(m^*)}}{x^{(M)}} \le 2$. The last expression can thus be further bounded by

$$\sum_{m=m^*}^{M} \left( -\frac{\log_2 \frac{x^{(m-1)}}{x^{(m)}}}{4(\ln T)\eta} \right) h \left( \frac{u_i}{x^{(m)}} \right) \le \sum_{m=m^*}^{M} \left( -\frac{\log_2 \frac{x^{(m-1)}}{x^{(m)}}}{4(\ln T)\eta} \right) h \left( \frac{u_i}{2x^{(M)}} \right)$$

$$= -\frac{\log_2 \frac{x^{(m^*-1)}}{x^{(M)}}}{4(\ln T)\eta} h \left( \frac{u_i}{2x^{(M)}} \right) \le -\frac{1}{4(\ln T)\eta} h \left( \frac{u_i}{2x^{(M)}} \right) \qquad (x^{(m^*-1)} \ge 2x^{(M)})$$

$$= -\frac{1}{4(\ln T)\eta} \left( \frac{u_i}{2x^{(M)}} - 1 - \ln \left( \frac{u_i}{2x^{(M)}} \right) \right)$$

$$= -\frac{1}{8(\ln T)\eta} \max_{t \in [T]} \frac{u_i}{x_{t,i}} + \mathcal{O} \left( \frac{\ln(NTu_i)}{\eta \ln T} \right) \le -\frac{1}{8(\ln T)\eta} \max_{t \in [T]} \frac{u_i}{x_{t,i}} + \mathcal{O} \left( \frac{1 + Nu_i}{\eta} \right),$$

where in the first inequality we use the fact for $m \ge m^*$, $\frac{u_i}{x^{(m)}} \ge \frac{u_i}{x^{(m^*)}} \ge \frac{u_i}{2x^{(M)}} \ge 1$, and that $h(y)$ is positive and increasing when $y \ge 1$.

On the other hand, if $x^{(M)} \ge \frac{1}{2N}$ or $x^{(M)} \ge \frac{u_i}{2}$, we have $\max_{t \in [T]} \frac{u_i}{x_{t,i}} = \frac{u_i}{x^{(M)}} \le 2Nu_i + 2$ and thus

$$\sum_{t=2}^{T} \left( \frac{1}{\eta_{t,i}} - \frac{1}{\eta_{t-1,i}} \right) h \left( \frac{u_i}{x_{t,i}} \right) \le 0 \le \frac{1}{8(\ln T)\eta} \left( -\max_{t \in [T]} \frac{u_i}{x_{t,i}} + 2Nu_i + 2 \right).$$

Considering both cases and Eq. (6), we get

$$\sum_{t=1}^{T} \left( D_{\psi_t}(u, x_t) - D_{\psi_t}(u, x_{t+1}) - \frac{\beta}{2} \sum_{t=1}^{T} \langle \nabla_t, u - x_t \rangle^2 \right)$$

$$\le \mathcal{O} \left( \beta N + \frac{N \ln T}{\eta} \right) + \frac{1}{8(\ln T)\eta} \sum_{i=1}^{N} \left( -\max_{t \in [T]} \frac{u_i}{x_{t,i}} + 2Nu_i + 2 \right) + \sum_{i=1}^{N} \mathcal{O} \left( \frac{1 + Nu_i}{\eta} \right)$$

$$\le \mathcal{O} \left( \frac{N \ln T}{\eta} \right) - \frac{1}{8(\ln T)\eta} \sum_{i=1}^{N} \max_{t \in [T]} \frac{u_i}{x_{t,i}},$$

finishing the proof. $\qquad \square$

**Lemma 7.** BARRONS *guarantees* $\sum_{t=1}^{T} \langle \nabla_t, x_t - x_{t+1} \rangle \le \frac{8N \ln T}{\beta}$.

*Proof.* Define $F_t(x) \triangleq \langle x, \nabla_t \rangle + D_{\psi_t}(x, x_t)$. Using Taylor's expansion and first order optimality of $x_{t+1}$ we have

$$F_t(x_t) - F_t(x_{t+1}) = \nabla F_t(x_{t+1})^\top (x_t - x_{t+1}) + \frac{1}{2}(x_t - x_{t+1})^\top \nabla^2 F_t(\xi_t)(x_t - x_{t+1})$$

$$\ge \frac{1}{2}(x_t - x_{t+1})^\top \nabla^2 F_t(\xi_t)(x_t - x_{t+1}) = \frac{1}{2} \|x_t - x_{t+1}\|^2_{\nabla^2 F_t(\xi_t)},$$

where $\xi_t$ is some point that lies on the line segment joining $x_t$ and $x_{t+1}$. On the other hand, by the definition of $F_t$, nonnegativity of Bregman divergence, and Hölder inequality, we have

$$F_t(x_t) - F_t(x_{t+1}) = \langle x_t - x_{t+1}, \nabla_t \rangle - D_{\psi_t}(x_{t+1}, x_t) \le \|x_t - x_{t+1}\|_{\nabla^2 F_t(\xi_t)} \|\nabla_t\|_{\nabla^{-2} F_t(\xi_t)}.$$

Combining the above two inequalities we get $\|x_t - x_{t+1}\|_{\nabla^2 F_t(\xi_t)} \le 2 \|\nabla_t\|_{\nabla^{-2} F_t(\xi_t)}$, and thus

$$\langle \nabla_t, x_t - x_{t+1} \rangle \le \|\nabla_t\|_{\nabla^{-2} F_t(\xi_t)} \|x_t - x_{t+1}\|_{\nabla^2 F_t(\xi_t)} \le 2 \|\nabla_t\|^2_{\nabla^{-2} F_t(\xi_t)}$$

$$= 2\nabla_t^T (\beta A_t + \nabla^2 \varphi_t(\xi_t))^{-1} \nabla_t \le \frac{2}{\beta} \nabla_t^\top A_t^{-1} \nabla_t,$$

where $\varphi_t$ is the log-barrier regularizer defined in the proof of Lemma 6 (whose Hessian is clearly positive semi-definite). Using Lemma 11 in [12], we continue with

$$\frac{2}{\beta} \sum_{t=1}^{T} \nabla_t^\top A_t^{-1} \nabla_t \le \frac{2}{\beta} \ln \frac{|A_T|}{|A_0|} \le \frac{2N \ln(1 + T^3)}{\beta} \le \frac{8N \ln T}{\beta},$$

where the second inequality uses the fact $\ln|A_0| = N \ln N$ and by AM-GM inequality $\ln|A_T| \le N \ln \frac{\mathbf{Tr}(A_T)}{N} \le N \ln \left( N + \frac{\sum_{t=1}^{T} \|\nabla_t\|_2^2}{N} \right) \le N \ln(N + NT^3)$ since $\|\nabla_t\|_2^2 \le N^2 T^2 (\sum_i r_{t,i}^2)/(\sum_i r_{t,i})^2 \le N^2 T^2$. This finishes the proof. $\qquad \square$

**Analysis of ADA-BARRONS** To prove Lemma 2, we make use of the following stability lemmas whose proofs are deferred to Appendix B.

**Lemma 8.** *In* ADA-BARRONS, *if* $\gamma \le \frac{1}{25}$, *then* $1 - \frac{\sqrt{\gamma}}{2} \le \frac{u_{t+1,i}}{u_{t,i}} \le 1 + \frac{\sqrt{\gamma}}{2}$ *for all $t$ and $i$.*

**Lemma 9.** *In* ADA-BARRONS, *if* $\eta \le \frac{1}{300}$, *then* $1 - \frac{\sqrt{3\eta}}{2} \le \frac{x_{t+1,i}}{x_{t,i}} \le 1 + \frac{\sqrt{3\eta}}{2}$ *for all $t$ and $i$.*

*Proof of Lemma 2.* Denote $a_t \triangleq \max_{s \in [t], i \in [N]} \frac{u_{t,i}}{x_{s,i}}$. Suppose $a_t$ attains its max at $s = s'$ and $i = i'$ (i.e., $a_t = \frac{u_{t,i'}}{x_{s',i'}}$), then when $s' \le t - 1$, we have by Lemma 8 $a_{t-1} \ge \frac{u_{t-1,i'}}{x_{s',i'}} = \left( \frac{u_{t-1,i'}}{u_{t,i'}} \right) a_t \ge \frac{a_t}{1 + \frac{\sqrt{\gamma}}{2}} \ge \frac{1}{2} a_t$; when $s' = t$, we have by Lemma 8 and 9 $a_{t-1} \ge \frac{u_{t-1,i'}}{x_{t-1,i'}} = \left( \frac{u_{t-1,i'}}{u_{t,i'}} \right) \left( \frac{x_{t,i'}}{x_{t-1,i'}} \right) a_t \ge \frac{1 - \frac{\sqrt{3\eta}}{2}}{1 + \frac{\sqrt{\gamma}}{2}} a_t \ge \frac{1}{2} a_t$. This concludes the proof. $\qquad \square$

*Proof of Lemma 3.* Define $\Gamma_t(u) = \sum_{s=1}^{t} f_s(x_s) - \sum_{s=1}^{t} f_s(u) - \frac{1}{\gamma} \sum_{i=1}^{N} \ln \frac{1}{u_i}$. Suppose condition (4) holds at some time $t$ at the end of epoch$(\beta)$ and cause the algorithm to restart. Then we know that $\beta \le \alpha_{t-1}(u_{t-1})$ and $\beta > \alpha_t(u_t)$. The first condition guarantees that Eq. (3) holds for $u_{t-1}$ at time $t - 1$. Also, note that $u_{t-1}$ is the maximizer of $\Gamma_{t-1}$. Together they imply for any $u \in \bar{\Delta}_N$,

$$\Gamma_{t-1}(u) \le \Gamma_{t-1}(u_{t-1}) \le \sum_{s=1}^{t-1} f_s(x_s) - \sum_{s=1}^{t-1} f_s(u_{t-1}) \le \mathcal{O}\left( \frac{N \ln T}{\eta} \right) + \frac{8N \ln T}{\beta} - \frac{a_{t-1}}{8(\ln T)\eta}, \tag{7}$$

where we recall the notation $a_t \triangleq \max_{s \in [t], i \in [N]} \frac{u_{t,i}}{x_{s,i}}$. The second condition implies

$$\frac{1}{\beta} < \frac{1}{\alpha_t(u_t)} = \max \left\{ 2, \max_{s \in [t]} 8|\nabla_s^\top(u_t - x_s)| \right\} = \max \left\{ 2, \max_{s \in [t]} 8 \left| \frac{\langle x_s, u_t - x_s \rangle}{\langle x_s, r_s \rangle} \right| \right\}$$

$$\le \max \left\{ 2, 8 \max_{s \in [t], i \in [N]} \frac{u_{t,i}}{x_{s,i}} + 8 \right\} = 8a_t + 8 \le 16a_{t-1} + 8, \tag{8}$$

where we apply Lemma 2 for the last step. Further combining this with Eq. (7), and noting that $f_t(x) - f_t(u) \le \max_i \ln \frac{u_i}{x_i} \le \ln(NT)$ for any $x \in \bar{\Delta}_N$, we have for any $u \in \bar{\Delta}_N$,

$$\sum_{s=1}^{t} f_s(x_s) - \sum_{s=1}^{t} f_s(u) \le \ln(NT) + \sum_{s=1}^{t-1} f_s(x_s) - \sum_{s=1}^{t-1} f_s(u) \le \ln(NT) + \Gamma_{t-1}(u) + \frac{N \ln(NT)}{\gamma}$$

$$\leq \mathcal{O}\left(\frac{N\ln T}{\eta}\right) + \frac{8N\ln T}{\beta} - \frac{1}{8(\ln T)\eta}\left(\frac{1}{16\beta} - \frac{1}{2}\right) \qquad \text{(by (7) and (8))}$$

$$\leq \mathcal{O}\left(N^2(\ln T)^3\right) - \frac{8N\ln T}{\beta}.$$

For the last epoch, we can apply Theorem 1 over the entire epoch and simply discard the negative term to obtain the claimed bound. □

## 5 Conclusions and Open Problems

We have shown that our new algorithm ADA-BARRONS achieves logarithmic regret of $\mathcal{O}(N^2(\ln T)^4)$ for online portfolio with much faster running time compared to Universal Portfolio, the only previous algorithm with truly logarithmic regret. A natural open problem is whether it is possible to further improve either the regret (from $N^2$ to $N$) or the computational efficiency without hurting the other. It is conjectured in [20] that FTRL with log-barrier [3] (i.e. our $u_t$'s) might also achieve logarithmic regret without dependence on $G$. On the pessimistic side, it is also a conjecture that it might be impossible to have the optimal regret with $\mathcal{O}(N)$ computations per round [19].

**Acknowledgements.** The authors would like to thank Tim van Erven for introducing the problem and to thank Tim van Erven, Dirk van der Hoeven, and Wouter Koolen for helpful discussions throughout the projects, especially on the FTRL approach. The work was done while KZ visited the University of Southern California. KZ gratefully acknowledges financial support from China Scholarship Council. HL and CYW are grateful for the support of NSF Grant #1755781.

## Footnotes

[1]Recent improvements on log-concave sampling such as [7, 16, 17] may lead to improved running time, but it is still a large polynomial.

[2]For online gradient descent, $G$ is the largest gradient $\ell_2$-norm.

[3]A recent work [6] uses a mixture of the Shannon entropy and log-barrier as the regularizer for a different problem.

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
