[Supplementary Material]

# A  Discussions on Tuning $\beta$

In this section, we discuss the challenges in tuning $\beta$ via other approaches. Recall that by the calculation shown after Theorem 1, a $\beta$ such that $\frac{1}{2}\alpha_T(u^*) \le \beta \le \alpha_T(u^*)$ where $u^* \triangleq \arg\min_{u \in \bar{\Delta}_N} \sum_{t=1}^{T} f_t(u)$ ensures a regret bound of $\mathcal{O}(N^2(\ln T)^3)$. We first show the existence of such $\beta$ when the environment is *oblivious*, that is, $r_1, \ldots, r_T$ are all fixed ahead of time. (However, we emphasize that our adaptive tuning method introduced in Section 3 does not rely on the existence of such $\beta$ at all and works even against non-oblivious environments.)

When $r_1, r_2, \ldots, r_T$ are fixed and thus $u^*$ is also fixed, one can view $\alpha_T(u^*)$ as a (complicated) function of $\beta$. It is not hard to see that this function is continuous: note that $x_{t+1}$ is a continuous function with respect to $\beta, A_t, \eta_t, x_t, \nabla_t$ because $x_{t+1}$ is the minimizer of a strongly convex function parameterized by these quantities. Also, $A_t, \eta_t, \nabla_t$ are continuous functions of $\{x_1, \ldots, x_t\}$.[4] So overall, $x_{t+1}$ is a continuous function of $\{\beta, x_1, \ldots, x_t\}$. By induction, we know that $x_t$ is a continuous function of $\beta$ for all $t$. Finally, since $\alpha_T(u^*)$ continuously depends on $\{x_1, \ldots, x_T\}$, it is also a continuous function of $\beta$.

Next note that the range of $\alpha_T(u^*)$ is $\left[\frac{1}{16NT}, \frac{1}{2}\right]$ because $8|\nabla_t^\top (u^* - x_t)| \le 8 \|\nabla_t\|_\infty \|u^* - x_t\|_1 \le 16NT$. Thus by intermediate value theorem, if we vary $\beta$ from $\frac{1}{32NT}$ to $\frac{1}{2}$, there must exist a $\beta$ such that $\frac{1}{2}\alpha_T(u^*) \le \beta \le \alpha_T(u^*)$, which completes our argument. In fact, by $\alpha_T(u^*)$'s continuity, the set of $\beta$'s satisfying the inequality will form an interval or a union of intervals.

Given that such $\beta$ does exist but is unknown, a natural idea is to instantiate $M$ copies of BARRONS's with different $\beta$'s forming a grid on $\left[\frac{1}{32NT}, \frac{1}{2}\right]$, then use Hedge to learn over these copies, which only introduces an additional $\ln M$ regret since the loss is exp-concave. If any of these $\beta$'s happens to fall into one of the intervals described above, then the algorithm has overall regret $\mathcal{O}(N^2(\ln T)^3 + \ln M)$.

However, the challenge is to figure out how dense the grid has to be, which depends on the slope (i.e. Lipschitzness) of $\alpha_T(u^*)$ with respect to $\beta$. The larger the slope, the denser the grid needs to be. Trivial analysis only shows that the Lipschitzness is exponential in $T$, which is far from satisfactory. Also note that the running time per round of this algorithm is $(MN^{3.5})$. Therefore even if $M$ is polynomial in $T$ which is good for the regret, it still defeats our purpose of deriving more efficient algorithms.

# B  Omitted Proofs

We first show that competing with smooth CRP from $\bar{\Delta}_N$ is enough.

**Lemma 10.** *For any $u' \in \Delta_N$, with $u = \left(1 - \frac{1}{T}\right) u' + \frac{1}{NT} \in \bar{\Delta}_N$ we have*

$$\sum_{t=1}^{T} f_t(x_t) - \sum_{t=1}^{T} f_t(u') \le \sum_{t=1}^{T} f_t(x_t) - \sum_{t=1}^{T} f_t(u) + 2.$$

*Proof.* By convexity of $f_t$, we have

$$\sum_{t=1}^{T} f_t(u) - \sum_{t=1}^{T} f_t(u') \le \sum_{t=1}^{T} \nabla f_t(u)^\top (u - u') \le \sum_{t=1}^{T} \frac{(u' - u)^\top r_t}{u^\top r_t}$$

$$\le \sum_{t=1}^{T} \frac{\left(\frac{u}{1-\frac{1}{T}} - u\right)^\top r_t}{u^\top r_t} = \frac{1}{1 - \frac{1}{T}} \le 2.$$

$\square$

Next we provide the omitted proofs for several lemmas.

*Proof of Lemma 5.* Note that the function $h_t(x) = e^{-2\beta f_t(x)} = \langle x, r_t \rangle^{2\beta}$ is concave since $0 \leq 2\beta \leq 1$. Therefore we have $h_t(u) \leq h_t(x_t) + \langle \nabla h_t(x_t), u - x_t \rangle$. Plugging in $\nabla h_t(x) = -2\beta e^{-2\beta f_t(x)} \nabla f_t(x)$ gives

$$e^{-2\beta f_t(u)} \leq e^{-2\beta f_t(x_t)} \left( 1 - 2\beta \langle \nabla_t, u - x_t \rangle \right),$$

or equivalently

$$f_t(u) \geq f_t(x_t) - \frac{1}{2\beta} \ln \left( 1 - 2\beta \langle \nabla_t, u - x_t \rangle \right)$$

By the condition on $\beta$ we also have $\left| 2\beta \langle \nabla_t, u - x_t \rangle \right| \leq \frac{1}{4}$. Using the fact $-\ln(1 - z) \geq z + \frac{1}{4}z^2$ for $|z| \leq \frac{1}{4}$ gives:

$$f_t(x_t) - f_t(u) \leq \langle \nabla_t, x_t - u \rangle - \frac{\beta}{2} \langle \nabla_t, x_t - u \rangle^2$$

$$= \langle \nabla_t, x_t - x_{t+1} \rangle + \langle \nabla_t, x_{t+1} - u \rangle - \frac{\beta}{2} \langle \nabla_t, x_t - u \rangle^2$$

$$\leq \langle \nabla_t, x_t - x_{t+1} \rangle + D_{\psi_t}(u, x_t) - D_{\psi_t}(u, x_{t+1}) - \frac{\beta}{2} \langle \nabla_t, x_t - u \rangle^2,$$

where the last step follows standard OMD analysis. More specifically, since $x_{t+1}$ is the minimizer of the function $F_t(x) \triangleq \langle \nabla_t, x \rangle + D_{\psi_t}(x, x_t)$, by the first-order optimality condition, we have $\langle u - x_{t+1}, \nabla F_t(x_{t+1}) \rangle \geq 0$ for all $u \in \bar{\Delta}_N$. Note $\nabla F_t(x_{t+1}) = \nabla_t + \nabla \psi_t(x_{t+1}) - \nabla \psi_t(x_t)$. Rearranging the condition gives $\langle \nabla_t, x_{t+1} - u \rangle \leq \langle \nabla \psi_t(x_{t+1}) - \nabla \psi_t(x_t), u - x_{t+1} \rangle$. Directly using the definition of Bregman divergence, one can verify $\langle \nabla \psi_t(x_{t+1}) - \nabla \psi_t(x_t), u - x_{t+1} \rangle = D_{\psi_t}(u, x_t) - D_{\psi_t}(u, x_{t+1}) - D_{\psi_t}(x_{t+1}, x_t)$, which is further bounded by $D_{\psi_t}(u, x_t) - D_{\psi_t}(u, x_{t+1})$ by the nonnegativity of Bregman divergence. This concludes the proof. $\qquad \square$

*Proof of Lemma 8.* Define $\Psi_t(u) = \sum_{s=1}^{t} f_s(u) + \frac{1}{\gamma} \sum_{i=1}^{N} \ln \frac{1}{u_i}$. We first show that if $\|u_t - u_{t+1}\|_{\nabla^2 \Psi_{t+1}(u_t)} \leq \frac{1}{2}$ holds, then the conclusion follows.

Indeed, note that $\nabla^2 \Psi_{t+1}(u_t) = \sum_{s=1}^{t+1} \frac{r_s r_s^\top}{\langle u_t, r_s \rangle^2} + \frac{1}{\gamma} \left[ \frac{1}{u_{t,i}^2} \right]_{\text{diag}} \succeq \frac{1}{\gamma} \left[ \frac{1}{u_{t,i}^2} \right]_{\text{diag}}$, where $\left[ \frac{1}{u_{t,i}^2} \right]_{\text{diag}}$ represents the $N$ dimensional diagonal matrix whose $i$-th diagonal element is $\frac{1}{u_{t,i}^2}$. We thus have

$$\|u_t - u_{t+1}\|_{\frac{1}{\gamma}[1/u_{t,i}^2]_{\text{diag}}} \leq \|u_t - u_{t+1}\|_{\nabla^2 \Psi_{t+1}(u_t)} \leq 1/2,$$

which implies $\frac{(u_{t,i} - u_{t+1,i})^2}{\gamma u_{t,i}^2} \leq \frac{1}{4}$, or $1 - \frac{\sqrt{\gamma}}{2} \leq \frac{u_{t+1,i}}{u_{t,i}} \leq 1 + \frac{\sqrt{\gamma}}{2}$ for all $i \in [N]$.

Next, we prove the inequality $\|u_t - u_{t+1}\|_{\nabla^2 \Psi_{t+1}(u_t)} \leq \frac{1}{2}$. Note $u_{t+1} = \operatorname{argmin}_{x \in \bar{\Delta}_N} \Psi_{t+1}(x)$. If we can prove $\Psi_{t+1}(u') > \Psi_{t+1}(u_t)$ for any $u'$ that satisfies $\|u' - u_t\|_{\nabla^2 \Psi_{t+1}(u_t)} = \frac{1}{2}$, then we obtain the desired inequality $\|u_t - u_{t+1}\|_{\nabla^2 \Psi_{t+1}(u_t)} \leq \frac{1}{2}$ by the convexity of $\Psi_{t+1}$.

By Taylor's expansion, we know there exists some $\xi$ in the line segment joining $u'$ and $u_t$, such that

$$\Psi_{t+1}(u') = \Psi_{t+1}(u_t) + \nabla \Psi_{t+1}(u_t)^\top (u' - u_t) + \frac{1}{2}(u' - u_t)^\top \nabla^2 \Psi_{t+1}(\xi)(u' - u_t)$$

$$= \Psi_{t+1}(u_t) + \nabla f_{t+1}(u_t)^\top (u' - u_t) + \nabla \Psi_t(u_t)^\top (u' - u_t) + \frac{1}{2} \|u' - u_t\|_{\nabla^2 \Psi_{t+1}(\xi)}^2$$

$$\geq \Psi_{t+1}(u_t) + \nabla f_{t+1}(u_t)^\top (u' - u_t) + \frac{1}{2} \|u' - u_t\|_{\nabla^2 \Psi_{t+1}(\xi)}^2$$

$$\geq \Psi_{t+1}(u_t) - \|\nabla f_{t+1}(u_t)\|_{\nabla^{-2} \Psi_{t+1}(u_t)} \|u' - u_t\|_{\nabla^2 \Psi_{t+1}(u_t)} + \frac{1}{2} \|u' - u_t\|_{\nabla^2 \Psi_{t+1}(\xi)}^2$$

$$= \Psi_{t+1}(u_t) - \frac{1}{2} \|\nabla f_{t+1}(u_t)\|_{\nabla^{-2} \Psi_{t+1}(u_t)} + \frac{1}{2} \|u' - u_t\|_{\nabla^2 \Psi_{t+1}(\xi)}^2 \qquad (9)$$

where the first inequality is by the optimality of $u_t$. As $\nabla^2 \Psi_{t+1}(u_t) \succeq \frac{1}{\gamma} \left[ \frac{1}{u_{t,i}^2} \right]_{\text{diag}}$ implies $\nabla^{-2} \Psi_{t+1}(u_t) \preceq \gamma \left[ u_{t,i}^2 \right]_{\text{diag}}$, we continue with

$$\|\nabla f_{t+1}(u_t)\|_{\nabla^{-2}\Psi_{t+1}(u_t)}^2 \leq \|\nabla f_{t+1}(u_t)\|_{\gamma\left[u_{t,i}^2\right]_{\text{diag}}}^2 = \frac{\gamma r_{t+1}^\top \left[u_{t,i}^2\right]_{\text{diag}} r_{t+1}}{\langle u_t, r_{t+1} \rangle^2} = \frac{\gamma \sum_{i=1}^N u_{t,i}^2 r_{t+1,i}^2}{\langle u_t, r_{t+1} \rangle^2} \leq \gamma.$$
(10)

Note $\xi$ is between $u_t$ and $u'$, so $\|\xi - u_t\|_{\nabla^2\Psi_{t+1}(u_t)} \leq \frac{1}{2}$ and thus $\frac{\xi_i}{u_{t,i}} \leq 1 + \frac{\sqrt{\gamma}}{2} \leqslant \frac{11}{10}$ according to previous discussions. Therefore, we have

$$\nabla^2 \Psi_{t+1}(\xi) = \sum_{s=1}^{t+1} \frac{r_s r_s^\top}{(r_s^\top \xi)^2} + \frac{1}{\gamma} \left[ \frac{1}{\xi_i^2} \right]_{\text{diag}} \succeq \frac{100}{121} \left( \sum_{s=1}^{t+1} \frac{r_s r_s^\top}{(r_s^\top u_t)^2} + \frac{1}{\gamma} \left[ \frac{1}{u_{t,i}^2} \right]_{\text{diag}} \right) = \frac{100}{121} \nabla^2 \Psi_{t+1}(u_t).$$
(11)

Now combining inequalities (9), (10) and (11), we arrive at

$$\begin{aligned}
\Psi_{t+1}(u') &\geq \Psi_{t+1}(u_t) - \frac{\sqrt{\gamma}}{2} + \frac{50}{121} \|u' - u_t\|_{\nabla^2\Psi_{t+1}(u_t)}^2 \\
&= \Psi_{t+1}(u_t) - \frac{\sqrt{\gamma}}{2} + \frac{25}{242} \\
&\geq \Psi_{t+1}(u_t),
\end{aligned}$$

which finishes the proof. $\qquad\square$

*Proof of Lemma 9.* The proof is similar to the proof of Lemma 8. Denote $F_t(x) = \langle x, \nabla_t \rangle + D_{\psi_t}(x, x_t)$. We again first prove that if $\|x_t - x_{t+1}\|_{\nabla^2 F_t(x_t)} \leq \frac{1}{2}$, then the conclusion follows.

Note $\nabla^2 F_t(x_t) = \beta A_t + \left[ \frac{1}{\eta_{t,i} x_{t,i}^2} \right]_{\text{diag}} \succeq \left[ \frac{1}{\eta_{t,i} x_{t,i}^2} \right]_{\text{diag}} \succeq \left[ \frac{1}{3\eta x_{t,i}^2} \right]_{\text{diag}}$, because $\eta_{t,i} \leq \eta \exp\left( \log_T(\frac{NT}{N}) \right) \leq 3\eta$. Thus we have

$$\|x_t - x_{t+1}\|_{\frac{1}{3\eta}[1/x_{t,i}^2]_{\text{diag}}} \leq \|x_t - x_{t+1}\|_{\nabla^2 F_t(x_t)} \leq 1/2,$$

which implies $\frac{(x_{t,i} - x_{t+1,i})^2}{3\eta x_{t,i}^2} \leq \frac{1}{4}$ and thus $1 - \frac{\sqrt{3\eta}}{2} \leq \frac{x_{t+1,i}}{x_{t,i}} \leq 1 + \frac{\sqrt{3\eta}}{2}$ for all $i \in [N]$.

It remains to prove the inequality $\|x_t - x_{t+1}\|_{\nabla^2 F_t(x_t)} \leq \frac{1}{2}$. Since $x_{t+1} = \operatorname{argmin}_{x \in \bar{\Delta}_N} F_t(x)$, if we can prove $F_t(x') > F_t(x_t)$ for all $x'$ that satisfies $\|x' - x_t\|_{\nabla^2 F_t(x_t)} = \frac{1}{2}$, then we obtain the desired inequality $\|x_t - x_{t+1}\|_{\nabla^2 F_t(x_t)} \leq \frac{1}{2}$ by the convexity of $F_t$. By Taylor's expansion, there exists some $\zeta$ on the line segment joining $x'$ and $x_t$, such that

$$\begin{aligned}
F_t(x') &= F_t(x_t) + \nabla F_t(x_t)^\top (x' - x_t) + \frac{1}{2}(x' - x_t)^\top \nabla^2 F_t(\zeta)(x' - x_t) \\
&= F_t(x_t) + \nabla_t^\top (x' - x_t) + \frac{1}{2} \|x' - x_t\|_{\nabla^2 F_t(\zeta)}^2 \\
&\geq F_t(x_t) - \|\nabla_t\|_{\nabla^{-2} F_t(x_t)} \|x' - x_t\|_{\nabla^2 F_t(x_t)} + \frac{1}{2} \|x' - x_t\|_{\nabla^2 F_t(\zeta)}^2 \\
&= F_t(x_t) - \frac{1}{2} \|\nabla_t\|_{\nabla^{-2} F_t(x_t)} + \frac{1}{2} \|x' - x_t\|_{\nabla^2 F_t(\zeta)}^2.
\end{aligned}$$
(12)

As $\nabla^2 F_t(x_t) = \beta A_t + \left[ \frac{1}{\eta_{t,i} x_{t,i}^2} \right]_{\text{diag}} \succeq \frac{1}{3\eta} \left[ \frac{1}{x_{t,i}^2} \right]_{\text{diag}}$, we have $\nabla^{-2} F_t(x_t) \preceq 3\eta \left[ x_{t,i}^2 \right]_{\text{diag}}$. Therefore

$$\|\nabla_t\|_{\nabla^{-2} F_t(x_t)}^2 \leq \|\nabla_t\|_{3\eta\left[x_{t,i}^2\right]_{\text{diag}}}^2 = \frac{3\eta r_t^\top \left[x_{t,i}^2\right]_{\text{diag}} r_t}{\langle x_t, r_t \rangle^2} = \frac{3\eta \sum_{i=1}^N x_{t,i}^2 r_{t,i}^2}{\langle x_t, r_t \rangle^2} \leq 3\eta.$$
(13)

Since $\zeta$ is between $x_t$ and $x'$, we have $\|\zeta - x_t\|_{\nabla^2 F_t(x_t)} < \frac{1}{2}$ and thus $\frac{\zeta_i}{x_{t,i}} \leq 1 + \frac{\sqrt{3\eta}}{2} \leq \frac{21}{20}$ according to previous discussions and the fact $\eta \leqslant \frac{1}{300}$. Therefore, we have

$$\nabla^2 F_t(\zeta) = \sum_{s=1}^{t} \frac{r_s r_s^\top}{(r_s^\top \zeta)^2} + \left[\frac{1}{\eta_{t,i}\zeta_i^2}\right]_{\text{diag}} \succeq \frac{400}{441}\left(\sum_{s=1}^{t} \frac{r_s r_s^\top}{(r_s^\top x_t)^2} + \left[\frac{1}{\eta_{t,i}x_{t,i}^2}\right]_{\text{diag}}\right) = \frac{400}{441}\nabla^2 F_t(x_t).$$

(14)

Now combining inequalities (12), (13) and (14), we get

$$\begin{aligned} F_t(x') &\geq F_t(x_t) - \frac{\sqrt{3\eta}}{2} + \frac{200}{441}\|x' - x_t\|_{\nabla^2 F_t(x_t)}^2 \\ &= F_t(x_t) - \frac{\sqrt{3\eta}}{2} + \frac{50}{441} \\ &\geq F_t(u_t), \end{aligned}$$

which finishes the proof. $\qquad\square$

## Footnotes

[4]The fact that $\eta_t$ is continuous with respect to $\{x_1, \ldots, x_t\}$ depends on our new increasing learning rate scheme and is not true for the scheme used in previous works [2, 21] based on doubling trick.