[Reviews · NeurIPS 2018]

Reviewer 1



The paper studies the online portfolio selection problem in which capital is divided between $N$ financial instruments and rebalanced in each round. The goal of the learner is to reweight her portfolio between rounds as to minimize her (multiplicative) regret compared to the best constant rebalanced portfolio. The problem is known to have an optimal regret of $O(N \log T)$ with $T$ being the time horizon that is implemented using a computationally-inefficient method. Many efficient implementations exist, most notably Online Newton Step which gives an $O(G N \log T)$ regret bound where $G$ is a problem-dependent parameter controlled by the adversary that can be made arbitarily large. This work presents a computationally-efficient algorithm that enjoys $O(N^2 \log^4 T)$ regret bound independently of $G$. The authors do so by augmenting the ONS algorithm with the logarithmic barrier such that each instrument has its own learning rate. By noticing that $G$ increases whenever the algorithm has a large regret against one of the instruments, the algorithm increases its learning rate to account for the increase in $G$ thus quickly canceling its effect on the regret. The paper is well written. The ideas are put forward in a clear and concise manner. Overall a very nice paper. Minor comments -------------- Line 87: tunning Line 216: should be $r_s$ instead of $x_s$

Reviewer 2



This paper introduces a new algorithm (BARRONS) for the online portfolio optimization problem. This algorithm obtains logarithmic regret in time quadratic in the number of rounds without any potentially catastrophic dependence on a gradient norm. This is in contrast to the Cover's universal portfolio algorithm, online newton step, and exponentiated gradient, which each achieve at most two of these three goals. The algorithm itself is an application of the online mirror descent framework, using the less-classical log-barrier regularizer. The paper provides good intuition for how the algorithm is able to avoid dependence on the gradient norm: the gradient norm is only big if a previously poorly-performing stock starts to perform well. As a result the learner has a kind of "surplus" of regret that it can fall back on while adapting to the new stock. The algorithm still leaves something to be desired: the total running time T^2N^(2.5) where N is the number of stocks (the dimension of the problem) and T is the number of rounds. Both of these dependencies are rather poor (although much better than the universal portfolio algorithm). Also the regret is N^2log^4(T), rather than the ideal N\log(T). The authors suggest that the T^2 aspect of the runtime might be improved by using more intelligent optimization algorithms in a subroutine of the main algorithm. This may not bear out, but it suggests that there are at least some obvious points of attack for improving the results. In a pleasant surprise for a theory paper, Most of a complete proof of the regret bound fits into the main text of the paper. Quality: This is reasonably clean approach to an old problem that achieves non-trivial new bounds. Overall this seemed a good-quality paper. Clarity: The background and the contrast between prior approaches was well-explained. Originality: The main original ideas in this paper seems to be the application of the log-barrier potential to this particular problem, and the intuition used to remove the gradient-norm dependence in the regret. I'm not aware of similar results on the problem. Significance: The main improvement of this paper is to remove the dependence on the gradient norm in the portfolio optimization problem. Both the regret bound and time complexity have some distressing high exponents, so there is still significantly more work to be done. However, the removal of the gradient norm from the bound is important as it gets rid of a kind of "wishful thinking" aspect of other bounds, and it is significantly faster than the only other algorithm to obtain this (universal portfolio, which is abysmally slow), so I think this paper represents a nice step of progress.

Reviewer 3



Summary: This paper considers the classical problem of online portfolios selection. Cover's algorithm achieves the optimal regret bound, but it is not computationally efficient. The authors propose variants of the Online Newton Step with a logarithmic barrier function as a regularizer. The regret bound is a bit inferior to that of Cover's, but still independent of the parameter G, which is an upper bound of norms of gradients of loss functions. The idea is to add a regularizer so that G is small enough w.r.t. N and T. Quality and Originality: The paper is technically sound and the idea is neat. But, the analysis follows the standard analysis of the FTRL. Significance: The paper is rather incremental than innovative. However, the technical improvement is non-trivial and interesting for the community.